# Analysis of Spatiotemporal Heterogeneity of Glacier Mass Balance on the Northern and Southern Slopes of the Central Tianshan Mountains, China

**Lin Liu [1,2,†], Hao Tian [1,†], Xueying Zhang [1], Hongjin Chen [1], Zhengyong Zhang [1,2,*] , Guining Zhao [1,*], Ziwei Kang [1], Tongxia Wang [1], Yu Gao [1], Fengchen Yu [1], Mingyu Zhang [1], Xin Yi [1] and Yu Cao [1]**

1   College of Sciences, Shihezi University, Shihezi 832000, China; liulin779@163.com (L.L.);
    haotian1996_china@163.com (H.T.); zxy970716@163.com (X.Z.); zlyzxlx@163.com (H.C.);
    kangziwei0808@163.com (Z.K.); wtx0428@163.com (T.W.); gyiks1225@163.com (Y.G.);
    yufengchen678@163.com (F.Y.); zmy_zz1015@163.com (M.Z.); jiuwuxin1331@126.com (X.Y.);
    cyfqq1230@163.com (Y.C.)
2   Key Laboratory of Oasis Town and Mountain-Basin System Ecology, Xinjiang Production and Construction
    Corps, Shihezi 832000, China
*   Correspondence: zyz0815@163.com (Z.Z.); 13345492733@163.com (G.Z.);
    Tel.: +86-18040830081 (Z.Z.); +86-13345492733 (G.Z.)
†   These authors contributed equally to this work.

**Abstract:** Glacier mass balance can visually indicate the degree of glacier response to climate change. The mountain glaciers are an essential source of recharge for rivers in arid regions and play a vital role in maintaining regional ecological stability and production life. This paper drives a spatially distributed degree-day model using multi-source remote sensing data such as MOD11C3 and TRMM3B43 to simulate the mass balance in the Tianshan Mountains' south and north slope basins. The spatiotemporal heterogeneity of the mass balance was compared and attributed using a Geographical detector. The results show that: (1) The glaciers in the north and south basins are mainly distributed at an altitude of 3900–4300 m, and the total glacier area accounts for 85.71%. The number of less than 1 km$^2$ glaciers is the most in the whole region. (2) During the study period, the glaciers in the north and south basins were negative (−465.95 mm w.e.) an entire interannual change rate was −28.36 mm w.e./a. The overall trend of ablation can be divided into two stages: from 2000 to 2010a, the persistence increased, and from 2010 to 2016a, the volatility decreased. (3) In the attribution of mass balance, the factors affecting glacier mass balance can be divided into two parts: climate and topography. The cumulative contribution rate of climate factors in Kaidu is nearly 20% higher than that of topographic factors, but the contribution rate of climate factors in Manas is only 7.3% higher. Therefore, the change of glacier mass balance in the Kaidu river basin is more driven by climate factors, while the glacier mass balance in the Manas river basin is more affected by the combination of climate and topographic factors. (4) The climate accumulation is the dominant factor in the Manas river basin (69.55%); for the ablation area, the Kaidu river basin is dominated by climate (70.85%), and the Manas river basin is dominated by topographic factors (54.11%). Due to the driving force of climate and topographic factors and the different coupling modes, glacier mass balance's spatiotemporal heterogeneity in the north and south slope basins is caused. This study contributes to analyzing the mechanism of regional changes in the glacier mass balance. It provides a scientific basis for investigating the characteristics of water resource changes and water resource regulation in the north and south slope basins of the Tianshan Mountains.

**Keywords:** Manas river basin; Kaidu river basin; glacier mass balance; Geographical detector; attribution analysis

## 1. Introduction

With global warming, as significant freshwater resources are known as "solid reservoirs", most glaciers are in a state of retreat and ablation. Some studies have also shown that mountain glaciers are more sensitive to climate warming [1,2]. At the same time, a series of disaster phenomena caused by the change and movement of mountain glaciers have also been widely valued by various research institutions [3]. As an indicator of climate change, glaciers mass balance is a critical junction connecting glaciers and climate, and it also comprehensively reflects the hydrothermal conditions of glacier development [4,5]. Glacier meltwater generated by the change of glacier mass balance has a stabilizing effect on regional water resources. Given the importance of glacier mass balance, it has been an important research content in glacier change detection worldwide. The World Glacier Monitoring Service (WGMS) aims to monitor global glacier changes continuously, systematically compile global glacier monitoring data, make long-term observations of 10 key reference glaciers, and publish their mass balance data every two years [6]. Urumqi glacier No.1, located in the Tianshan Mountains in Central Asia, is one of the reference glaciers included in the ten essential monitoring items of WGMS, which can reflect the basic characteristics of the changes in central Asian glaciers to a certain extent [7,8]. However, affected by topography and regional climate conditions, the monitoring data of a single glacier cannot comprehensively reflect the overall change characteristics of regional glaciers. Selecting suitable regions to reflect the spatial heterogeneity of glacier mass balance changes can further explain the internal factors affecting the changes in the mass balance of mountain glaciers, which will be more conducive to understanding the mechanism of their recharge to rivers.

The research methods about glacier mass balance mainly include field observation, remote sensing monitoring, model simulation, etc. [9–11], glacier surface observation can provide detailed parameters. Still, it is difficult to reflect the spatial heterogeneity of the region due to the significant human interference in the observation process, which is usually confined to a single glacier in some areas. In addition, remote sensing monitoring has the advantages of a wide range and multiple time series. For example, the target decomposition technology using polarimetric SAR data has broad application prospects in ice research. Polarimetric decompositions have become an effective method for identifying glacier areas, especially over debris-free glaciers [12–14]. The above techniques can realize the accurate measurement and detection of single or multiple glaciers. Still, it is difficult to reflect the spatial heterogeneity of glacier change in a large area [7]. Compared with glacier observation and remote sensing monitoring, model simulation can better reflect the physical process mechanism of mass balance [15,16]. It can be divided into the empirical model based on the degree-day factor and the physical model based on energy balance. Compared with glacier observation and remote sensing monitoring, model simulation can better reflect the physical process mechanism of mass balance [17] and can be divided into empirical models based on degree-day factors and physical models based on energy balance. The energy balance model is based on the energy balance principle, considers the energy input and output items, and focuses more on the mechanism of the mass balance process. However, the input parameters and variables are complex and challenging to obtain. There are high requirements for data accuracy or resolution, which limits the popularization and application of the model. The degree-day model is simple and easy to obtain, its results are accurate and reliable in large-scale glacier studies, and it is most widely used. By comparing and analyzing numerous models and methods, the degree-day model regards the degree-day factor as a statistical variable of albedo, latent sensible heat flux, relative humidity, and other factors. The degree-day model adopts a constant representation according to the difference in time and space, which effectively reduces the model's complexity. Therefore, if a large-scale glacier mass balance study is to be carried out, taking into account factors such as topography and climate, and taking into account the advantages of the simplicity and universality of the degree-day model and balance of

the ice melt/snowmelt module, a degree-day model based on the water balance principle is more suitable for the model. [18].

As an ecological barrier of an oasis in the arid area of northwest China, the dynamic change of glacier mass balance in Tianshan Mountain is of great scientific value in assessing the impact of climate on water resources [9]. The Tianshan Mountains in China are 250–350 km wide from north to south. The most extensive independent zonal mountain system globally, the farthest mountain system from the ocean globally, and the most major mountain system in the arid region of the world. At present, the study of glacier mass balance in The Tianshan Mountains of China is mainly limited to the analysis of measured data of a single glacier, the simulation of glacier mass balance in different regions or the whole basin, and the impact of climate change on regional mass balance [19–22]. There are few studies on the difference and comparison of glacier mass balance in the same mountain system under different site conditions. The regional differences in ecological and geographical characteristics on the northern and southern slopes of the Tianshan Mountains affect the development and evolution of glaciers in different regions, leading to varying factors of glacier mass balance changes in the different areas. Comparison and attribution analysis of the mass balance of glaciers on the north and south slopes will make a comparison to explain further the mechanism of the change of mountain glaciers' mass balance. In this paper, glaciers in the Manas river basin on the northern slope of the central Tianshan Mountains and the Kaidu river basin on the southern slope as research objects. Using MOD11C3 and TRMM3B43 as driving data, a spatially distributed degree-day model was constructed to analyze and compare the mass balance of glaciers in typical watersheds on the north and south slopes of the Geographical detector to make an attribution analysis of their spatiotemporal heterogeneity characteristics. This study is helpful to analyze the mechanism of regional changes in mountain glacier mass balance and provides the scientific basis for exploring the features of water resource change and water resources regulation in the northern and southern slopes of the Tianshan Mountains.

## 2. Study Area

Tianshan Mountain, located in the hinterland of Eurasia, is the most extensive independent latitudinal mountain system in the world, which stretches for more than 1700 m in China. It crosses the central part of Xinjiang Uygur Autonomous Region and is the boundary between northern and southern Xinjiang [23]. It consists of parallel mountain ranges divided into north, middle, and south zones. The study area is located in the central Tianshan Mountains, including the Yilianhabierga Mountains, Saarming Mountains, Bortoula Mountains, etc. Yilianhabierga Mountains are about 320 km long in the east and west, 110 km wide in the north and south, and the mountains are generally above 4000 m above sea level [24]. The Manas River on the northern slope and the Kaidu River on the southern slope selected in this study are derived from Mount Ylianhabirga (Figure 1). The south slope of Tianshan Mountain has less precipitation, is deep inland, and has an arid climate. The north slope of Tianshan Mountain has more precipitation than the southern slope because of the humid airflow from the Arctic Ocean and the Atlantic Ocean. The climate is relatively moist. The southern slope has sufficient sunlight, but the northern slope is less, and the temperature of the south slope is higher than that of the north slope. Manas river basin, located on the north slope of the central Tianshans Mountain, is the basin with the most significant number and scale of glaciers in the inland area of Junggar and has a moderate temperate continental arid climate [25,26]. The Kaidu River, located on the southern slope of the central Tianshan Mountains, is one of the four primary sources of the Tarim River Basin, the longest inland river in China, and belongs to the warm temperate continental arid climate [25,26].

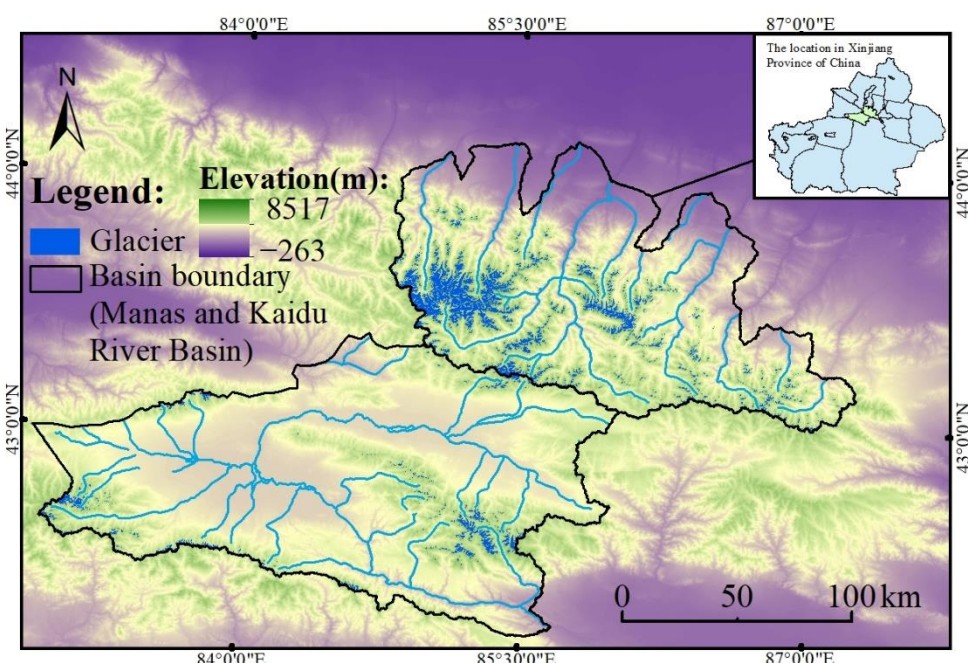

**Figure 1.** Location of the study area.

## 3. Data and Methods

The data of this paper are divided into multi-source remote sensing data, glacier area, and boundary vector data of the Manas river basin and the Kaidu river basin. Remote sensing raster data include DEM, MOD11C3, TRMM3B43, $\geq 0$ °C accumulated temperature, precipitation, air temperature, solar radiation, etc. (Table 1). The specific applications of the data in this paper are as follows: DEM can reflect the topographic characteristics of mountainous areas and can be used to obtain topographic factors in mountainous regions, which is the basis for the spatial discretization of various factors of glacier mass balance to construct degree-day model, and it is also a factor that affects the change of glacier mass balance. MOD11C3 and TRMM3B43 are monthly data from 2001 to 2016a, the primary meteorological data driving the distributed degree-day model. The climatic background data such as $\geq 0$ °C accumulated temperature, precipitation, air temperature, and solar radiation are put into the Geographical detector and the terrain factors as climate factors to explore the reasons for the changes in the glacier mass balance. Among them, air temperature and precipitation are also important indicators to construct the ice/snow degree-day factor. The glacier area vector data come from the Second China Glacier Catalog Dataset.

**Table 1.** Data sources.

| Type of Data | Resolution | Data Sources (Accessed on 17 November 2021) |
|---|---|---|
| DEM | 30 m × 30 m | RESDC (https://www.resdc.cn/data.aspx?DATAID=217) |
| MOD11C3 | 0.05° × 0.05° | NASA (https://lpdaac.usgs.gov/products/mod11c3v006/) |
| TRMM3B43 | 0.25° × 0.25° | NASA (https://disc.gsfc.nasa.gov/datasets/TRMM_3B43_7/) |
| Solar radiation | 10 km × 10 km | TPDC (https://data.tpdc.ac.cn/) |
| $\geq 0$ °C accumulated temperature Air temperature Precipitation | 0.5 km × 0.5 km | RESDC (https://www.resdc.cn/DOI/doi.aspx?DOIid=39) |
| The Second China Glacier Catalog Dataset | — | NCDC (http://www.ncdc.ac.cn/) |

*3.1. Simulated Glacier Mass Balance*

MODIS and TRMM have an effective spatiotemporal resolution. They have a better objective and scientific description of the mountainous areas with few measuring stations. MODIS and TRMM are involved in the climate data inversion of the Himalayas and Alps, such as using MOD10A1 and MYD10A1 to study the snow line change in the Alps [27]; MODIS land surface temperature data are also used to retrieve the temperature of the Qinghai Tibet Plateau and the central Himalayas [28,29]. In the relevant studies using TRMM, most of them focus on the spatial distribution characteristics of precipitation and the accuracy of precipitation data [30,31]. The above research should prove the feasibility of MODIS and TRMM in describing temperature and precipitation in mountainous areas and has a high reference for the study.

This paper used the distributed degree-day model to simulate the mass balance of glaciers in typical basins on the north and south slope of the central Tianshan Mountains, to analyze the temporal and spatial differences in glacier mass balance between the Kaidu river basin and Manas river basin and drive the geographic detector with climate and topographic factors to explore the leading factors affecting mass balance change in both basins and make a comparative analysis. The methodology flowchart is as follows (Figure 2):

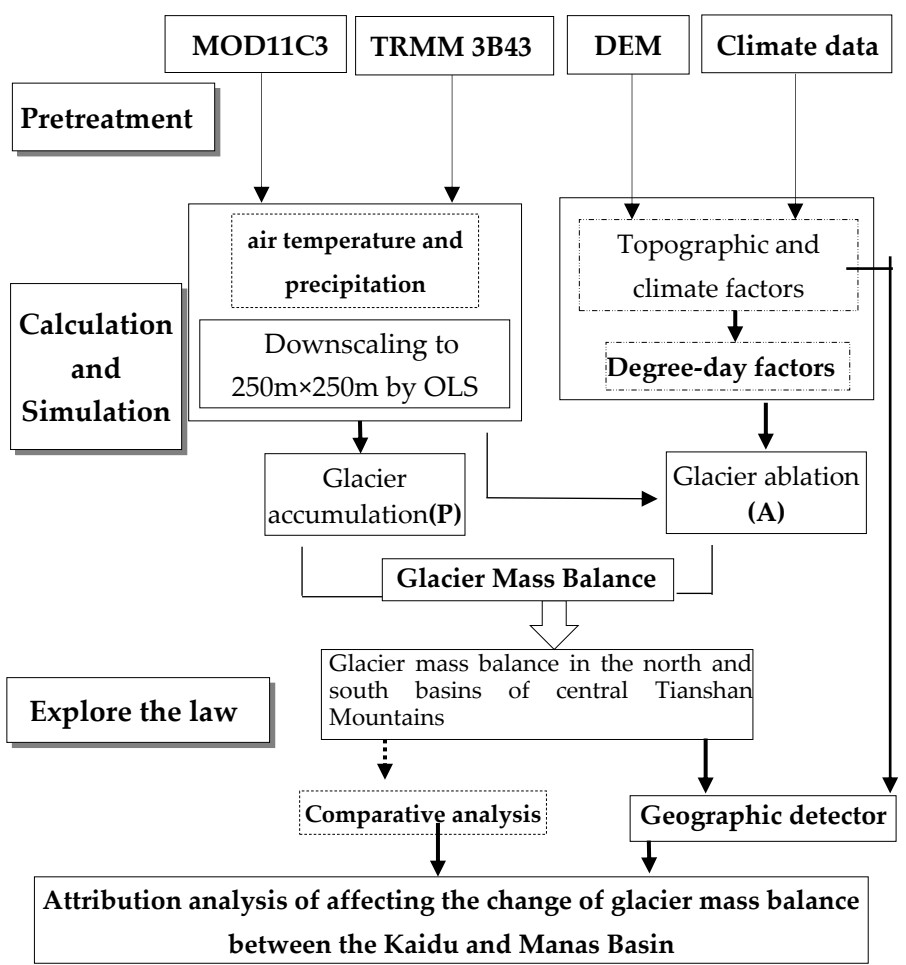

**Figure 2.** Methodology flowchart.

The degree-day model simplifies the process mechanism of glacier ablation and accumulation. Still, the traditional degree-day model is only a conceptual model. In contrast, the distributed model improves the ability to express the spatial characteristics of each element in the degree-day model, enhances its physical basis, and divides the glacier area into grids. The spatial variation of the overall glacial mass balance is illustrated by describing the

specific variation of each grid cell in terms of a set of factors that affect the variation of the glacial mass balance. This way shows the spatial change of the entire glacier mass balance [32]. Due to the different resolutions of the multi-source remote sensing data driving the distributed degree-day model, it is difficult to describe the ice surface climate with too low resolution. Therefore, in this study, the OLS model was constructed in the process of modeling the distributed degree-day model to downscale MOD11C3 and TRMM3B43 to 250 m × 250 m to meet the model accuracy requirements. In contrast, the distributed degree-day model reflects the spatial variation characteristics of the mass balance through the grid.

For the ablation of glacier ice and snow, the amount of ablation in a certain period is expressed as follows [15,16,33,34]:

$$A = D \cdot PDD \tag{1}$$

where $A$ is the ablation of water equivalent of glacier ice and snow in a certain period (mm w.e.). $D$ is the degree-day factor of glacier ice/snow (mm d$^{-1}$ °C$^{-1}$). $PDD$ is the positive accumulated temperature in a certain period. It can be obtained by the following formula [35]:

$$PDD = \int_{N_1-1}^{N_2} \frac{1}{\delta\sqrt{2\pi}} \int_0^{+\infty} T_m e^{\frac{-(T_m - T_a)^2}{2\delta^2}} dTdt \tag{2}$$

where it is assumed that the monthly average temperature (°C) within the year is usually distributed, $Ta$ is the annual average temperature (°C), $\delta$ is the standard deviation of the temperature distribution, $N_1$, $N_2$ are the start and the end dates of the calculation, and the period is $N = N_2 - N_1 + 1$.

The net mass balance at a point (or height) is [24]:

$$Bi = P + A + f \tag{3}$$

where $Bi$ is the glacier mass balance in a certain period (mm w.e.). $P$ is the accumulation amount of the glacier surface in a certain period, that is, the solid precipitation (mm). $A$ is the amount of ablation. $f$ is the amount of meltwater infiltration and freezing or internal supply (mm w.e.), usually calculated as 10% of the ablation volume.

The solid precipitation can be calculated according to the limiting temperature method [36]:

$$P_s = \begin{cases} P & T \le T_S \\ \frac{T_L - T}{T_L - T_S} P & T_S < T < T_L \\ 0 & T \ge T_L \end{cases} \tag{4}$$

$$P_L = P - P_S \tag{5}$$

where $P_s$ and $P_L$ are the solid and liquid precipitation (mm), $P$ is the total monthly precipitation (mm). $T$ is the monthly average temperature (°C), $T_S$ and $T_L$ are the limiting temperatures of solid and liquid precipitation (°C). Usually, it can be determined according to the observation results of Urumqi glacier No.1 in the Tianshan Mountains and related literature, and take $-0.5$ °C and $2$ °C, respectively.

### 3.2. Attribution of Glacier Mass Balance Change

The Geographical detector can effectively explain the spatial coupling between geographic phenomena and multiple factors [2,37,38]. In this paper, the factor detection module of the Geographical detector is used to quantitatively detect whether the factors

affect the spatial heterogeneity of the changes in the glacier mass balance in the study area and determine the degree of influence of each factor. The formula is as follows:

$$q = 1 - \frac{\sum\limits_{h=1}^{L} N_h \sigma_h^2}{N\sigma^2} = 1 - \frac{SSW}{SST} \tag{6}$$

where $h = 1, 2, \ldots, L$ is the stratification of the variable $Y$ or factor $X$, the classification or partition. $N_h$ and $N$ are the number of units in the $h$ stratum and the whole area, respectively. $\sigma_h^2$ and $\sigma^2$ are the variances of the stratum $h$ and the whole area $Y$. $SSW$ and $SST$ are the sums of the conflict within the layer and the total friction of the whole area, respectively.

The factors affecting the change of glacier mass balance are diverse, and the internal mechanisms are complex, but they can be roughly classified into two major factors: topography and climate. Topographic factors mainly include altitude, slope, slope aspect, etc., and climatic factors mainly comprise air temperature, precipitation, accumulated temperature, solar radiation, etc. [2,39–41]. Each factor is coupled and influenced by each other, which affects the changes in the glacier mass balance. In particular, the hydrothermal conditions and terrain complexity of the north and south slopes of the Tianshan Mountains will further cause spatial differences in the glacier mass balance on the north and south slopes [42,43]. Through the factor detection of the Geographical detector, the $q$-value of the above factors in each geographical area is calculated and sorted. The higher the $q$-value is, the stronger the explanatory power of the factor for the change of glacier mass balance is. According to the contribution rate, the driving factors of glacier mass balance change in northern and southern slopes can be identified and attributed.

## 4. Results

### 4.1. Analysis of Glacier Area in the Central Tianshan Mountains

Glacier area data are derived from the Second China Glacier Catalog Dataset, which is used to determine the distribution of glaciers in each basin during the mass balance calculation, and also as the basis for calculating the glaciers mass balance of individual glaciers. In order to know the distribution of glacier's area in the vertical direction, all the glaciers in both basins were arranged according to their average elevation. Most of the large glaciers are located at high elevations and are concentrated. Still, the number is generally small, while the small glaciers are found at low elevations but are widely distributed and numerous (Figure 3). At the same time, glacier area is the most intuitive manifestation of glacier size. The glacier size of each elevation zone can also be known by counting the glacier area in each elevation zone, which influences the change of mass balance. When the glacier size is large, it can effectively restrain the glacier melting. While the glacier size is small [44], it is easier to melt. The lowest elevation range of glaciers' distribution in both basins is 3500–3700 m, the highest is 4500–4800 m, and most of the large-scale glaciers are in the range of 3900–4300 m. The total area of glaciers at 4100–4300 m is the largest, about 560 km$^2$, the 3900–4100 m is the second elevation range, and the area at 3500–3700 m is the least, which is less than 20 km$^2$. The glaciers in the Kaidu river basin are mainly concentrated in the southwest and northeast of the basin. In contrast, the glaciers in the Manas river basin are focused on the southern part of the basin. The number of glaciers in the Manas river basin is 2.7 times that of the Kaidu river basin. The maximum area of a single glacier in the Manas river basin is 36 km$^2$, while that in the Kaidu river basin is 16 km$^2$. Overall, the total volume of glaciers in the Manas river basin is more extensive than those in the Kaidu river basin.

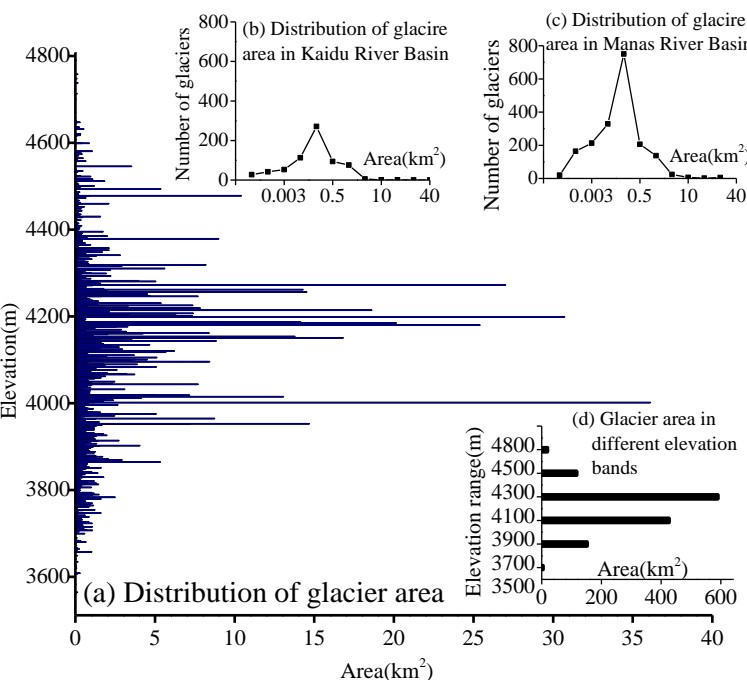

**Figure 3.** Elevation distribution characteristics of a single glacier area in both basins of central Tianshan Mountains. (**a**) Distribution of glacier area; (**b**) Distribution of glacier area in Kaidu River Basin; (**c**) Distribution of glacier area in Manas River Basin; (**d**) Glacier area in different elevation bands.

Generally speaking, small glaciers are the most sensitive to climate change, and their change of mass balance is the most pronounced. Therefore, the glaciers in both basins are divided into 11 classes according to the area size, with 0.001, 0.003, 0.005, 0.1, 0.5, 1, 5, 10, 15, 20, and 40 km². Although there are some differences in the number of glacier bars between both basins, glacier size distribution is very similar. There are a lot of glaciers in both basins, which are smaller than 1 km², with peak numbers occurring between 0.01 and 0.03 km². At the same time, it can be found that the larger area of glaciers in both basins, the fewer number with an area of more than 10 km². There is no glacier with an area of more than 20 km² in the Kaidu river basin. In addition, the small glaciers are more likely to melt because of their isolation and distribution in low altitudes. So the key to maintaining the mass balance of glaciers in both basins is the existence of a small number of medium and large area glaciers. The cooling effect on the surrounding environment due to the large size of the glacier can have a mitigating effect on glacier melt.

*4.2. Comparison of Glacier Mass Balance between Both Basins in the Central Tianshan Mountains*

Based on China's Second Glacier Cataloguing data, the mass balance of a single glacier in the Manas river basin and the Kaidu river basin was compared and analyzed (Figure 4). Overall, the mass balance of glaciers in the central Tianshan Mountains is mainly distributed between 0–1000 mm w.e., and the number of glaciers in −1000–1500 mm w.e. is significantly lower than former, and the number of glaciers between −2000 mm w.e. and 0 mm w.e. is the least. It can be seen that the mass balance of the glaciers in the central Tianshan Mountains is dominated by negative balance, and the distribution characteristics are similar to the normal distribution. The glaciers in a positive balance in the Kaidu river basin are significantly less than those in the Manas river basin. Still, the mass balance of most glaciers in the Kaidu river basin is between 0 and −500 mm w.e., while the mass balance of most glaciers in the Manas river basin is between 0 and −1000 mm w.e. There are more glaciers in the Manas river basin with a mass balance of over −1500 mm w.e. than in the Kaidu river basin. The maximum value of the mass balance of glaciers in the Kaidu river basin is comparable to that of the Manas river basin. Although the glaciers' mass

balance spans in both basins are similar, the Manas river basin has more glaciers and a higher negative balance.

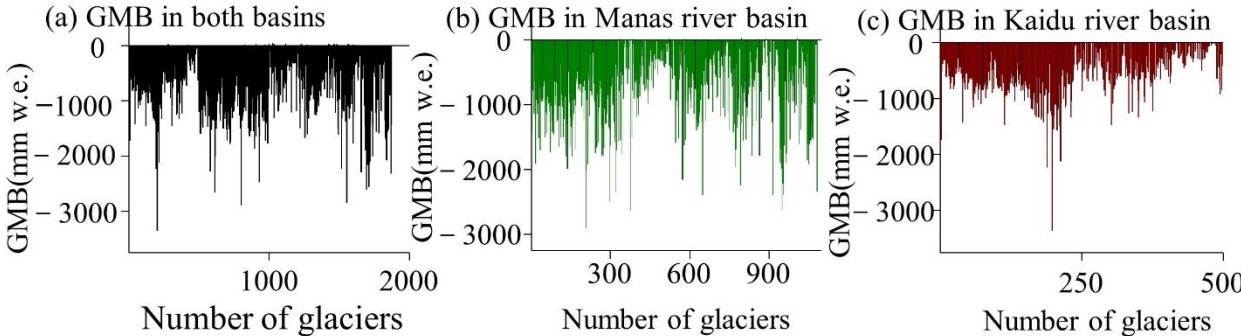

**Figure 4.** The mass balance of a single glacier in both basins of the central Tianshan Mountains. (**a**) GMB in both basins; (**b**) GMB in Manas river basin; (**c**) GMB in Kaidu river basin.

The interannual changes in the mass balance of glaciers can show the overall changes in the watershed and more clearly reflect the regional changes in mass balance (Figure 4). During the study period, the glaciers in both basins were in a negative balance state and continued to melt (−465.95 mm w.e.), with an interannual variation rate of −28.36 mm w.e./a. The most obvious glacier ablation was in 2010, followed by 2014; The melting trend can be divided into two stages: 2001–2010a is a continuous increase, and 2010–2016a is weakened volatility. Comparing both basins, from 2001 to 2016, the glacier ablation in the Kaidu river basin was 3.79% higher than that in the Manas river basin, but the change rate of the glacial mass balance in the Manas river basin (−28.85 mm w.e./a) slightly higher than the Kaidu tiver basin (−26.34 mm w.e./a). Glacier ablation in both basins reached the highest value in 2010, and the ablation in the Kaidu river basin (878.89 mm w.e.) was higher than that in the Manas river basin (811.26 mm w.e.). The trajectories of mass balance in both basins are similar with time. Still, the volatility of the trajectory in the Manas river basin is higher than that of the Kaidu river basin, with only a 1.07 mm w.e. difference. According to its spatial distribution map (Figure 5), the high value of glacier increased mass balance in both basins appears at the glacier aggregation region. The low mass balance values are mainly distributed at the end of glaciers or in clusters of small glaciers. The melting risk of dispersed glaciers is higher than that of glaciers with a high aggregation degree. It is not difficult to find that the glaciers on the southern slope are smaller in size. Still, the average ablation volume is higher than that on the northern slope, and the glaciers on the north slope are larger in scale but have higher ablation rates than those on the southern slope. This regional difference is closely related to the hydrothermal conditions and topographic conditions of the northern and southern slopes of the central Tianshan Mountains.

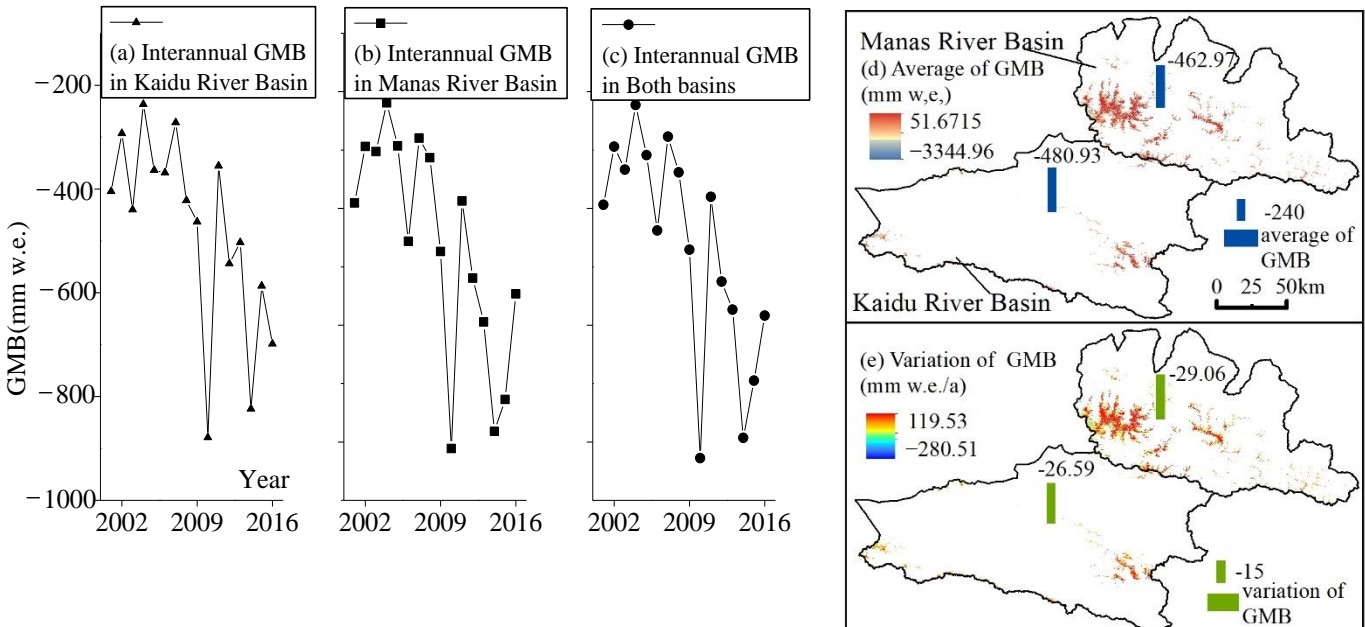

**Figure 5.** Glacier mass balance in both basins in central Tianshan (2001–2016). (**a**) Interannual GMB in Kaidu River Basin; (**b**) Interannual GMB in Manas River Basin; (**c**) Interannual GMB in Both Basin; (**d**) Average of GMB; (**e**) Variation of GMB.

In mainly glacier research, the Tianshan glaciers, as the representative of continental glaciers, play a pivotal role. Many scholars have carried out many types of research on changes in the mass balance of some glaciers in various basins in the Tianshan Mountains. Among them, Wucheng, X. et al. estimated the mass balance of the Tianshan glaciers in China based on the Landsat and ICESat data from 2003 to 2009a. They concluded that the average mass balance of the Tianshan Mountains during the period was −120 mm w.e. [45]. However, their estimated results were small compared with this study. Xiaohui, H. et al. used the geodetic method to estimate that the average mass balance of glaciers in the Tu-Haalik area from 1972 to 2016a was 180 mm w.e. Still, during the period from 1999 to 2016a, it reached 360 mm w.e. [46], which is far greater than the previous value of glacier mass balance, demonstrating the accelerated loss of glaciers since 2000. Ninglian, W. et al. collected the latest results of recent research on the mass balance of glaciers in the Tianshan Mountains. They found that the average mass balance of the Junggar basin was 600 mm w.e. [47], which was slightly larger than that of the present study. In addition, Puyu, W. et al. estimated that from 1972 to 2011a, glacier No. 6 in Yushugou, Miaoergou, and No. 1 glacier in Urumqi Heyuan in the East Tianshan Mountains were −510 and −510 mm w.e. [48]. Yanjun, C. et al. obtained the average mass balance of the Qingbingtan No. 72 glacier in the South Tianshan Mountains from 2008 to 2014a to be −510, −510, and −380 mm w.e., respectively [49], which are similar to the results of this study.

*4.3. The Attribution of Glacier Mass Balance Change in the North–South Slope Basin in the Central Tianshan Mountains*

By using factor detection and interaction detection based on a Geographical detector, the dominant factor affecting the mass balance can be determined by ranking the factor *q*-value (Table 2). The detector found that the dominant factors affecting the change of mass balance in the basin can be classified into topography and climate. Altitude is an important factor affecting the mass balance in the study area, followed by >0 °C accumulated temperature and annual average temperature. The contribution of annual average precipitation to solar radiation is equivalent. Topographic relief, slope, and aspect have limited influence on mass balance change. The contribution rate of topographic factors to both basins reached 44.87%, slightly lower than that of climatic factors (51.27%).

Therefore, the climate factor is still the dominant factor affecting the change of glacier mass balance in both basins. Although independent climatic factors have less influence than altitude, the interaction and coupling of each factor will have a stronger effect on glacier mass balance change.

**Table 2.** Contribution rate of mass balance factor affecting glacier (%).

| Basins | Elevation | Topographic Relief | Slope | Aspect | Solar Radiation | Air Temperature | Precipitation | >0 °C Accumulated Temperature |
|---|---|---|---|---|---|---|---|---|
| Both Basins | 42.75 | 1.06 | 1.07 | 3.85 | 11.08 | 14.41 | 10.14 | 15.64 |
| Kaidu | 30.88 | 0.77 | 0.79 | 2.09 | 14.73 | 11.31 | 26.90 | 12.53 |
| Manas | 40.44 | 1.02 | 1.02 | 3.85 | 11.00 | 14.70 | 8.02 | 19.95 |

The most critical factor affecting mass balance in the Kaidu river basin is altitude, followed by precipitation and its contribution rate is equivalent to elevation, reaching 26.9%. It shows that the precipitation factors greatly influence the glacier mass balance in the Kaidu river basin. The contribution of solar radiation, air temperature, and accumulated temperature is not much different. The contribution rate of slope direction is slightly higher than that of topographic relief and slope, but the contribution rate of the three is the low value of the whole factors. The contribution rate of topographic relief to the slope is less than 1% in the Kaidu. The contribution rate of factors affecting glacier mass balance in the Manas is higher than 1%. The first factor is still altitude, with the contribution rate of 40.45%, followed by the accumulated temperature of >0 °C. The contribution rate of solar radiation to temperature is equivalent, and the precipitation's contribution rate is slightly lower than the first two factors. Similar to the Kaidu river basin, the contribution rate of topographic relief, slope, and aspect are also low. Still, the contribution rate of the three in the Manas river basin is slightly higher than that in the Kaidu river basin, and the contribution rate of the slope is 3.85%. In the overall role of topographic factors and climatic factors, the Kaidu and Manas river basins are the same, still dominated by climatic factors. However, in the Kaidu river basin, the contribution rate of the climate factor is nearly 20% higher than that of the topographic factors. The Manas river basin's climate factors contribution rate is only 7.3% higher than the topographic factors, which is closely related to the more complex topographic environment in the Manas river basin. As the Manas river basin is located on the northern slope of the Tianshan Mountains, the precipitation is relatively abundant (annual average precipitation is 257.5 mm, Figure 6), reducing the warming effect of solar radiation on glaciers. The average elevation in the basin is 4094 m. The large glaciers are primarily distributed at high altitudes, which is more conducive to their accumulation, alleviating the Manas river basin's glacier ablation. There are few large glaciers in the Kaidu river basin, mainly distributed as small glaciers, and the average elevation of a single glacier in the basin is 3984 m (Figure 6). Glacier positive equilibrium is limited. Although the precipitation is less than in the Manas river basin, precipitation is liquid mainly at low altitudes, so it is difficult to stay on small glaciers. Higher heat in water droplets can accelerate the melting of small glaciers and increase their negative balance. According to the calculation, the ice climate data of both basins are obtained, and the average annual temperature of the glacier in the Manas river basin is −13.1 °C. In comparison, the Kaidu river basin is −9.7 °C. The main reason is that the Kaidu river basin is located on the southern slope of the Tianshan Mountains. Its solar radiation is higher than that of the Manas river basin, with a value of 179.29 W/m². The accumulated temperature above 0 °C is 337.6 °C, which is significantly higher than the Manas river basin (182.5 °C) on the northern slope of the Tianshan Mountains (Figure 6). Under the combined effect of topography and climate factors, the negative balance intensity of the Kaidu river basin is higher than that of the Manas river basin.

Under the dual influence of climatic factors and topographic factors, the glaciers in both basins of the central Tianshan Mountains are melting. Still, there are some differences

in the dominant factors affecting the melting of glaciers on the northern and southern slopes. The terrain on the south slope of the Tianshan Mountains is relatively flat and has a less natural place for glacier development. The limited precipitation cannot effectively accumulate glaciers at high altitudes. Temperature and solar radiation are higher than those on the northern slope, which is more likely to promote glacier ablation. Glacier ablation is more than on the north slope. Due to the complex terrain and more large glaciers on the north slope, such conditions effectively alleviate the glacier's negative balance, and thermal energy mainly affects ablation factors. The precipitation affects large glaciers or upstream glaciers in the basin, but the influence ability is limited.

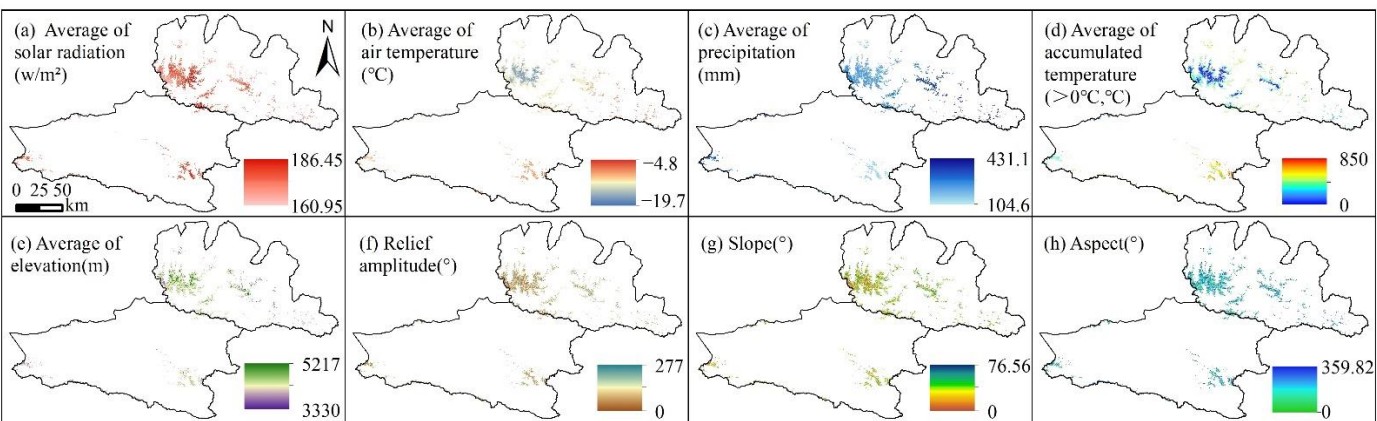

**Figure 6.** Driving factors affecting glacier mass balance. (**a**) Average of solar radiation (w/m$^2$); (**b**) Average of air temperature (°C); (**c**) Average of precipitation (mm); (**d**) Average of accumulated temperature (>0 °C, °C); (**e**) Average of elevation (m); (**f**) Relief amplitude (°); (**g**) Slope (°); (**h**) Aspect (°).

## 5. Discussion

The changes in the mass balance of glaciers are affected by various influencing factors. These factors will also produce corresponding eigenvalues in mass balance changes, and different degrees of mass balance will be adapted to other climatic and topographical conditions. If the attribution comparison is based only on the mean value of the mass balance of glaciers in the region, the attribution analysis of glacier accumulation and ablation (positive and negative mass balance) in the basin is very likely to be ignored. This paper takes the positive and negative balance as the breakthrough point, combining it with the factor detection of the Geographical detector and comparing the differences between both basins. The mechanisms for glacial mass balance changes on the north and south slopes of the central Tianshan Mountains can be further refined. The glaciers were divided into positive and negative balance areas based on the balance value by distributed degree-day model. Using the Geographical detector to detect the contribution rate of each factor. The driving force of the positive and negative balance areas in both basins will be further discussed.

### 5.1. Contribution Analysis of Influencing Factors Based on Positive and Negative Mass Balance Area

The part in positive balance is considered the accumulation area, and the part in negative balance is regarded as the ablation area based on the value of glacier mass balance. From the results of the Geographical detector (Figure 7), the factors affecting the accumulation of glaciers in the two basins are dominated by climatic factors. The cumulative contribution rate of climate factors is 69.37%, of which the precipitation contribution rate is the highest, followed by the contribution rate of temperature. The contribution rate of solar radiation is only 3.09%, and the effect of solar radiation on the accumulation of glaciers is minimal. Among topographic factors, altitude still has a high contribution rate (26.53%), while the contributions of the other three topographic factors are all low (Table 3). For glacier accumulation areas, hydrothermal conditions are critical factors. Mountains

provide a basic place for glaciers to accumulate/develop. It is possible to have a certain accumulation effect on glaciers. In the context of climate warming, the continuous melting of glaciers is a significant trend; the accumulation area of glaciers is mainly concentrated on the main body of glaciers with higher altitudes and larger scale, and they have the characteristics of seasonal accumulation [50–52]. The climate factor is still dominant (52.2%), but its cumulative contribution rate is equal to the topographic factor (47.8%). Among the climatic factors, the precipitation factor has the highest contribution rate (17.73%), followed by the solar radiation factor (14.85%). Among the topographic factors, the altitude contribution rate is the highest, which is 38.32%. The remaining three topographic factors have a low contribution rate to the ablation area. Different from the accumulation area, the elevation factor is the factor with the highest contribution to the negative balance of glaciers in the whole basin.

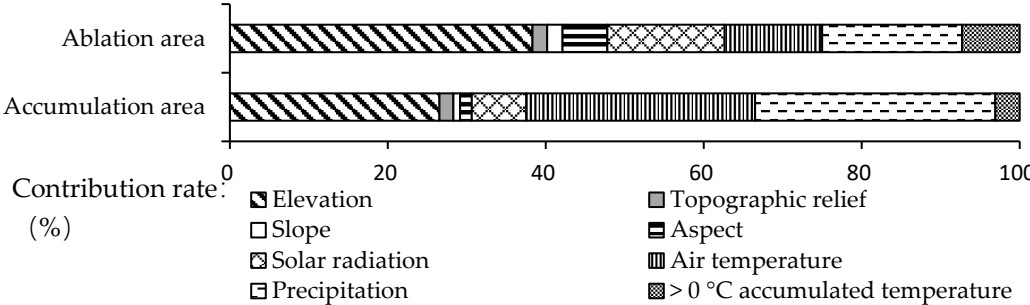

**Figure 7.** The contribution rate of each factor in the accumulation/ablation area.

**Table 3.** Each factor's statistical value in the accumulation/ablation area in the central Tianshan Mountains.

| Area | GMB (mm/w.e.) | Elevation (m) | Topographic Relief (°) | Slope (°) | Aspect (°) | Solar Radiation (W/m²) | Air Temperature (°C) | Precipitation (mm) | >0 °C Accumulated Temperature (°C) |
|---|---|---|---|---|---|---|---|---|---|
| Accumulation | 30.21 | 4502.18 | 60.19 | 37.67 | 128.92 | 157.41 | −14.16 | 241.94 | 128.84 |
| Ablation | −718.97 | 3980.93 | 37.64 | 25.34 | 187.71 | 175.67 | −11.45 | 254.88 | 260.91 |

Under the comprehensive action of multiple factors, there are significant differences in the natural environment in the positive and negative balance areas of the north and south slopes of the central Tianshan Mountains. According to statistics, the accumulation area is 33.3 km², the ablation area is 860.13 km², and the mass balance of the ablation area is dominant (Table 3). The topographic complexity of the accumulation area is higher than that of the ablation area, and the high altitude directly weakens the temperature of the environment in the accumulation area. The complex topographic relief is very likely to form the mountain's shadow to block the sunlight, thus eliminating the direct effect of solar radiation on the glacier surface and reducing the potential risk of glacier endothermic melting [53,54]. As shown in Table 3, the temperature and precipitation in the accumulation area are significantly lower than those in the ablation area. Most accumulation areas face the southeast, while the ablation areas are mostly facing the south. Therefore, the solar radiation received by the accumulation area is lower than that of the ablation area. The ablation area is mainly located in the lower altitude area, and the end of the glacier is straightforward to be ablated. The terrain here is relatively flat and more susceptible to solar radiation. Although the average temperature for many years is −11.45 °C, the accumulated temperature >0 °C is significantly higher than that in the accumulation area. It also has a higher risk of melting and is characterized by significant seasonal ablation.

*5.2. Attribution Comparison of Both Basins Based on Positive and Negative Balance Area*

The geographical environment of the Manas river basin on the northern slope of the Tianshan Mountains is quite different from that of the Kaidu river basin on the southern slope. The formation of positive and negative balance areas in the basins are also different, which can be further revealed through comparative analysis of the internal mechanism of the spatial heterogeneity of the mass balance of Tianshan glaciers. Topographic factors dominate the accumulation area and ablation area in the Kaidu river basin, and the ablation area is dominated by climatic factors, with cumulative contributions of 55.2% and 70.84%, respectively. The accumulation area in the Manas river basin is dominated by climatic factors, with a cumulative contribution rate of 69.38%, and the ablation area is dominated by topographical factors, with a cumulative contribution rate of 54.1% (Figure 8). The natural conditions of both basins control the patterns of glacier mass balance in the basins so that the dominant factors are distinctively regional.

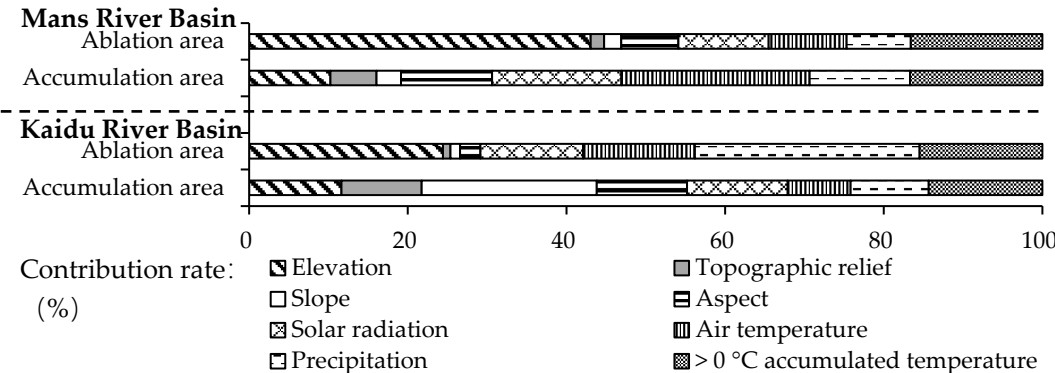

**Figure 8.** The contribution rate of each factor in the accumulation/ablation area in both basins.

In the accumulation area, the Manas river basin's positive mass balance and topographic complexity are higher than the Kaidu river basin (Table 4). It is worth noting that the slope factor has a very high explanatory power (22.06%) in the accumulation area of the Kaidu river basin, which may be related to the easy retention of solid snowfall on gentle slopes [55,56]. The accumulation area in the Manas river basin has the highest contribution rate of air temperature (23.73%). Taking into account factors such as more glaciers and larger scales in the Manas river basin, and then comparing climate data such as temperature and precipitation, it can be inferred that the stability of glaciers in the Manas river basin in the accumulation zone is higher and the surrounding environmental conditions are more conducive to the formation of positive mass balance of glaciers than those in Kaidu river basin. In the ablation area, the difference between both basins in the statistics value of the factors is slight. Still, the negative mass balance value of the Manas river basin is higher. The highest contribution rate of precipitation factor in the Kaidu river basin (28.33%) was slightly higher than that of elevation. The highest contribution rate of elevation factor in the Manas river basin (43.05 %). Glacier ablation is partly due to seasonal liquid precipitation at low altitudes [57–59]. The mean elevation of the ablation area in the Manas river basin is similar to that of the Kaidu river basin. The altitude range of about 3900–4000 m can be seen as a strong ablation zone in both basins. Although the dominant factors of the ablation/accumulation areas between both basins are different, in the final analysis, the interaction and coupling between topographic and climatic factors cause the differences in the mass balance of the glaciers in the basins.

In this study, comparing the differences in mass balance in the basins failed to go to a more detailed time scale. It focused on discussing natural environmental factors on the mass balance of glaciers between both basins without considering the direct or indirect effects of human activities on glaciers.

**Table 4.** Statistical value in the accumulation/ablation area of both basins.

| Basins | GMB (mm/w.e.) | Elevation (m) | Topographic Relief (°) | Slope (°) | Aspect (°) | Solar Radiation (W/m²) | Temperature (°C) | Precipitation (mm) | >0 °C Accumulated Temperature (°C) |
|---|---|---|---|---|---|---|---|---|---|
| | | | | Accumulation Area | | | | | |
| Kaidu | 20.37 | 4346.73 | 52.94 | 33.90 | 135.01 | 170.08 | −9.16 | 138.38 | 400.76 |
| Manas | 31.18 | 4517.40 | 60.90 | 38.04 | 128.33 | 167.11 | −14.65 | 252.08 | 102.21 |
| | | | | Ablation Area | | | | | |
| Kaidu | −591.73 | 3944.43 | 36.80 | 24.79 | 184.22 | 178.11 | −9.78 | 219.20 | 333.47 |
| Manas | −769.77 | 4006.34 | 38.06 | 25.62 | 189.70 | 174.71 | −12.12 | 268.97 | 231.87 |

## 6. Conclusions

This paper uses distributed degree-day model driven by multi-source remote sensing data. It simulates the glacier mass balance of the south and north basins of the Tianshan Mountains from 2001 to 2016. Based on the analysis and comparison of the temporal and spatial changes, using eight factors, including terrain factors and climate factors, were introduced by a Geographical detector to conduct an attribution comparison from the basin scale and the positive and negative mass balance area scale in the basin. The conclusions are as follows:

(1) In terms of glacier distribution, the glaciers in both basins are mainly distributed in the altitude range of 3500–4800 m. Most glaciers are distributed in 3900–4300 m, and the total area in this altitude range accounts for 85.71%. Due to the altitude and mountain trend, the glaciers in Kaidu are mainly concentrated in the southwest and northeast, which are mainly concentrated in the south of Manas. In terms of glacier scale, the number of less than 1 km² glaciers is the most, and the number of glaciers over 10 km² is rare; in terms of glacier quantity and scale, the Manas river basin is more than the Kaidu river basin.

(2) During the study period, the glaciers in both basins were continuously melting. They were in negative balance (−465.95 mm w.e.), and the interannual change rate was −28.36 mm w.e./a. among them, the glacier melting was the most obvious in 2010. The melting trend can be divided into two stages: the persistence increased from 2000 to 2010a, and the volatility decreased from 2010 to 2016a. Compared with both basins, the amount of glacier melting in Kaidu is 3.79% higher than that in Manas, but the change rate of glacier mass balance in Manas (−28.85 mm w.e./a) is slightly higher than that in Kaidu (−26.34 mm w.e./a).

(3) In the attribution of mass balance, the factors affecting glacier mass balance can be divided into climate and topography. The total contribution rate of topographic factors is 44.87%, which is slightly lower than that of climatic factors (51.27%). Therefore, climatic factors are still the dominant factors affecting the difference in glacier mass balance between both basins; however, in Kaidu, the contribution rate of climatic factors is nearly 20% higher than that of topographic factors. The contribution rate of climate factors in Manas is only 7.3% higher, so the change of glacier mass balance in Kaidu is more driven by climate factors, while the glacier mass balance in Manas is more affected by the combination of climate and topographic factors.

(4) The factors affecting the positive/negative mass balance of glaciers in both basins are dominated by climate factors (69.37%, 52.2%). Still, the contribution gap between climate and terrain factors in the ablation area is smaller. In terms of accumulation area, the positive mass balance in Kaidu is dominated by topographic factors (55.2%), while that in Manas is dominated by climatic factors (69.37%). For the ablation area, climate factors are dominated in Kaidu (70.85%), and topographic factors are dominated in Manas (54.11%). The driving forces of climate and terrain factors and the different coupling modes lead to the spatiotemporal heterogeneity of glacier mass balance in the north and south slope basins in the central Tianshan Mountain.

**Author Contributions:** Conceptualization, L.L. and H.T.; Investigation, X.Z. and H.C.; Methodology, Z.Z. and G.Z.; Resources, X.Y. and Y.C.; Software, M.Z.; Validation, Y.G. and F.Y.; Visualization, Z.K. and T.W.; Writing—original draft, H.T.; Writing—review and editing, L.L. and H.T. All authors have read and agreed to the published version of the manuscript.

**Funding:** This research was supported by The National Natural Science Foundation of China (Grant No. 41761108, 41641003), The third Comprehensive Scientific investigation project in Xinjiang (Grant No. 2021xjkk08).

**Data Availability Statement:** Not applicable.

**Acknowledgments:** We acknowledge the research environment provided by Xinjiang Production and Construction Corps Key Laboratory of Oasis Town and Mountain-basin System Ecology.

**Conflicts of Interest:** The authors declare no conflict of interest.

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
