# Peer review of "Analysis of Spatiotemporal Heterogeneity of Glacier Mass Balance on the Northern and Southern Slopes of the Central Tianshan Mountains, China"

_water, doi:10.3390/w14101601_

Round 1

Reviewer 1 Report

The reviewer would like to thank the authors for this work. The manuscript is informative and useful to the domain of hydrology. The authors are encouraged to put in some additional work to further improve its quality. 

Introduction

The authors mention the impact of climate change on the water resource in the alpine environment. In this regards the authors are requested to cite and highlight the contribution of the following article reporting a massive glacier disaster over the Himalayas and the impact of such event on the glacier mass balance.

Shugar et al., “A massive rock and ice avalanche caused the 2021 disaster at Chamoli, Indian Himalaya” Science, 2021.

Data

The authors have only considered the MOD and TRMM data. Target decomposition techniques using polarimetric SAR data have promising potential in cryospheric research. Polarimetric decompositions are an effective way to observe the electromagnetic scattering diversity and identification of glacier zones, particularly over debris-free glaciers. The authors are requested to indicate the importance of such techniques highlighted in the following articles and cite them in the manuscript. 

Touzi, IEEE TGRS, 2007. “Target scattering decomposition in terms of roll-invariant target parameters.”

Muhuri et al., IEEE JSTARS, 2017. “Scattering mechanism-based snow cover mapping using Radarsat-2 C-band polarimetric SAR data.” 

Bhattacharya et al., IEEE JSTARS, 2015. “Modifying the Yamaguchi four-component decomposition scattering powers using a stochastic distance.”  

Figs. and Tables

The way the authors have presented the tables and figures is not satisfactory. The authors are requested to present them in a better manner. Please include high-resolution, enlarged figures.

Conclusion

The authors are requested to enumerate their key contributions. At the moment the section is not well organized.

Reviewer 2 Report

Dear authors, 

I particularly have issues with the datasets you have used in this study. Kindly provide authentic published work from various regions of the World such as Alps and Himalayas that have used similar datasets for this type of work.

The datasets that have been used in this study, I don't think can be used to map glaciers and calculate the mass balance.

There are various other issues in the manuscript, that I have annotated in the attached pdf.

Round 2

Reviewer 2 Report

Thank you very much for answering all the queries. Best of luck